# Comparison of Characteristics and Survival Rates of Resectable Pancreatic Ductal Adenocarcinoma according to Tumor Location

**DOI:** 10.3390/biomedicines9111706

**Published:** 2021-11-17

**Authors:** Min Kyu Sung, Yejong Park, Bong Jun Kwak, Eunsung Jun, Woohyung Lee, Ki Byung Song, Jae Hoon Lee, Dae Wook Hwang, Song Cheol Kim

**Affiliations:** 1Division of Hepato-Biliary and Pancreatic Surgery, Department of Surgery, University of Ulsan College of Medicine and Asan Medical Center, 88, Olympic-Ro 43-Gil, Songpa-Gu, Seoul 05505, Korea; skystar8608@naver.com (M.K.S.); blackpig856@gmail.com (Y.P.); bjkwak0827@gmail.com (B.J.K.); eunsungjun@amc.seoul.kr (E.J.); ywhnet@gmail.com (W.L.); mtsong21c@naver.com (K.B.S.); gooddr23@naver.com (J.H.L.); drdwhwang@gmail.com (D.W.H.); 2Convergence Medicine, University of Ulsan College of Medicine, Seoul 05505, Korea

**Keywords:** pancreatic ductal adenocarcinoma, location, head, body/tail, survival

## Abstract

The impact of tumor location on patient survival in pancreatic ductal adenocarcinoma (PDAC) remains controversial. This study investigated the association between primary tumor location and survival rates for resectable PDAC. Additionally, we assessed if this association remains consistent across categories of the Tumor-Node-Metastasis staging system. We analyzed 2471 patients who underwent surgical resection between 2000 and 2018 at a single center. Subgroup analysis was performed according to the Tumor-Node-Metastasis staging system. Among the group, 67.9% (1677 patients) had pancreatic head cancer (PHC) and 32.1% (794 patients) had pancreatic body/tail cancer (PBTC). Patients with PHC had worse overall survival and worse disease-free survival than those with PBTC. Patients with PHC had worse survival in stage IB and stage IIB than those with PBTC. No significant difference was observed for stages IA, IIA, and III. Multivariate analysis showed that elevated CA 19-9, mGPS, a longer hospital stay, complication, accompanying vein resection, larger tumor size, worse differentiation, higher TNM stage (stage IIB, III, IV), presence of LVI, and positive resection margin were risk factors for poor survival after resection. In resectable PDAC, patients with PHC had worse overall and disease-free survival than those with PBTC. However, tumor location was not an independent prognostic factor for PDAC.

## 1. Introduction

Pancreatic ductal adenocarcinoma (PDAC) is the leading cause of cancer-related deaths worldwide, and its annual incidence and prevalence are steadily increasing [1,2]. Owing to improvements in diagnostic modalities and frequent health screening, early diagnosis of PDAC has become possible, which has increased overall survival rates [3]. Nevertheless, the mortality rate for PDAC remains exceptionally high due to its rapid progression and quick recurrence [3]. Therefore, it is important to understand factors that can influence PDAC prognosis.

PDAC is categorized into pancreatic head cancer (PHC) and pancreatic body and tail cancer (PBTC) based on anatomical location and embryologic origin [3]. Since PHC and PBTC have different clinical symptoms and treatment regimens, a significant difference between their survival rates has been previously reported [3]. There were several studies that have suggested the anatomical location of the PDAC is a prognostic factor for survival [4,5,6,7,8,9,10,11]. However, recently several studies suggested the opposite results [12,13,14,15]. In general, survival rates and prognosis are worse in patients with PBTC compared with those with PHC due to late diagnosis in the absence of noticeable symptoms such as obstructive jaundice [4,9,13,16,17,18,19]. However, few studies have shown similar postoperative oncological outcomes for PHC and PBTC despite larger tumor sizes among PBTC patients [5,7]. In contrast, some studies have shown the survival for PHC was worse than that for PBTC, which included early stage resectable PDAC [20,21].

Therefore, the impact of tumor location on patient survival for PDAC has been controversial, and the association seems to vary according to tumor characteristics such as size and stage. In this study, we investigated the association between primary tumor location and overall survival rates among resectable PDAC patients. Furthermore, we investigated whether this association remains consistent across categories of the Tumor-Node-Metastasis (TNM) staging system. Moreover, we determined the prognostic factors for overall survival for both PHC and PBTC patients.

## 2. Materials and Methods

### 2.1. Study Participants

All PDAC patients who underwent surgical resection between January 2000 and December 2018 at a single center were consecutively enrolled in this retrospective cohort study. Patients who underwent any operations for palliative measures (i.e., bypass surgery, open and biopsy) were excluded. Only patients diagnosed with PDAC were enrolled in this study. The number of patients diagnosed with other histological carcinomas (such as adenosquamous carcinoma, signet ring cell carcinoma, colloid carcinoma, and so on) was very small and they generally had a poorer prognosis than PDAC. Therefore, we excluded those with other histological diagnoses. Among the 2730 PDAC patients, we excluded 26 patients because their tumor locations could not be classified as either PHC or PBTC due to large tumor size. We excluded an additional 233 patients who underwent neoadjuvant chemotherapy due to variations in their chemotherapeutic regimen. For resectable pancreatic cancer, surgery is performed first, followed by adjuvant chemotherapy. However, for borderline resectable pancreatic cancer, neoadjuvant chemotherapy is recommended before surgery. Further, in these cases, some patients may have a higher T stage compared to others. If they are included in the resectable pancreatic cancer group, there may be a bias in the survival rate analysis. Therefore, these patients were excluded from this study. Ultimately, 2471 patients were included in the final analysis. Among the 2471 PDAC patients, 1677 (67.9%) patients had PHC, and 794 (32.1%) patients had PBTC.

### 2.2. Preoperative Evaluation and Surgical Procedure

Participants underwent preoperative, pancreas protocol computed tomography (CT) and magnetic resonance cholangiopancreatography (MRCP) for the initial evaluation of cancer stage and resectability. If distant metastasis was suspected, participants underwent 18F-fluorodeoxyglucose positron emission tomography (FDG-PET). We performed upfront surgery on patients who met the criteria for resectable tumors. If pancreatic cancer was confined to the pancreatic parenchyma, a standard procedure based on anatomical location was performed (pancreaticoduodenectomy for PHC and distal pancreatectomy for PBTC). If the resection margin was positive from the frozen biopsy, we performed a total pancreatectomy for complete R0 resection. Vascular resection was performed when PDAC had invaded the adjacent vessels, and organ-combined resection was performed when PDAC had invaded adjacent organs. Since 2006, we have performed laparoscopic surgery in cases of benign or suspected early PDAC. There have been 24 cases of robot-assisted surgery since 2016.

### 2.3. Data Collection

Patient data were collected from their electronic medical records. The data included demographic factors, namely age, sex, and body mass index (BMI), and American Society of Anesthesiologists (ASA) scores as well as preoperative carbohydrate antigen (CA 19-9), carcinoembryonic antigen (CEA), C reactive protein (CRP), and albumin levels to calculate the modified Glasgow prognostic score (mGPS) [22,23]. Length of hospital stay was categorized as ≤15 days and >15 days. We evaluated the presence of postoperative pancreatic fistula (POPF) and its complications. POPF was defined according to the 2016 International Study Group of Pancreas Surgery, and surgery-related complications were graded according to the Clavien–Dindo system [24,25]. We further categorized the participants according to adjuvant chemotherapy and chemoradiation therapy status. Tumors were categorized according to location as either head or body/tail cancer, and the operation method was categorized as either open or laparoscopic/robotic surgery. We also evaluated the status of the vein resection, artery resection and organ combined resection. Pathologic characteristics such as tumor size, differentiation, TNM staging, lymphovascular invasion (LVI), perineural invasion (PNI), and resection margin were evaluated. The degree of tumor differentiation was described according to the World Health Organization’s nomenclature, and TNM staging was classified based on the American Joint Committee on Cancer (AJCC) cancer staging system (8th edition) [26].

### 2.4. Outcome

The primary outcome was the overall survival (OS) rates for both the PHC and PBTC groups. Patients were followed up from the date of operation to the date of last outpatient clinic visit or death, whichever came first. The secondary outcome was disease-free survival (DFS) rates for both groups. DFS was defined as the time to relapse or all-cause death, whichever came first. Recurrence was defined as recurrence noted from a follow-up CT scan regardless of cancer-related symptoms. Of the 2471 patients, 1859 patients had an unambiguous date of death due to recurrence of PDAC after surgery. The exact date of death of 612 patients is unknown. Survival analysis was performed based on the date of the last outpatient visit. There were no deaths due to factors other than PDAC during this period.

### 2.5. Statistical Analysis

Descriptive analysis was performed for baseline patient characteristics. Categorical variables were presented as number and percentage, and continuous variables were presented as either mean (standard deviation) or median (interquartile range). The Chi-square test was used to compare categorical variables, and the Student’s *t*-test was used to compare continuous variables. We plotted the Kaplan–Meier survival curves and performed a log-rank test to compare OS and DFS rates of the PHC and PBTC groups. To determine prognostic factors for survival, the Cox proportional hazard model was used to calculate the relative risk ratios (RRs) and 95% confidence intervals (CIs). Subgroup analysis was performed according to the TNM stage. A two-tailed *p* < 0.05 was considered statistically significant, and all analyses were conducted using SPSS version 23.0 (IBM Corp., Armonk, NY, USA).

## 3. Results

### 3.1. Comparison of Patient Characteristics in PHC and PBTC Group

Table 1 and Table 2 show the comparison of demographic, clinical, surgical, and pathological patient characteristics of the PHC and PBTC groups. Among the 2471 PDAC patients, 1677 (67.9%) patients had PHC, and 794 (32.1%) patients had PBTC. The prevalence of obesity was higher in PBTC patients. The PHC group had a higher proportion of patients with elevated CA 19-9, had higher mGPS scores, and had patients with longer length of hospital stay compared with the PBTC group. Clinically relevant POPF were more prevalent in PBTC patients. Complications above Grade B or C were higher in PBTC patients but there was no significance. They had undergone a greater number of laparoscopic and robot-assisted surgeries. The incidence of vein resection was higher in PHC patients, whereas the incidence of organ combined resection was higher in PBTC patients. PBTC patients also had larger tumor sizes and more distant metastasis, whereas PHC patients had more lymph node metastasis, LVI and PNI.

### 3.2. Comparison of Survival Rates for PHC and PBTC Patients

The median survival for the entire cohort was 27 months (range 0–227 months). Figure 1 shows the OS and DFS curves according to the location of PDAC. Both OS and DFS rates were lower among PHC patients compared with PBTC patients (median overall survival: 24 months vs. 34 months, *p* = 0.001; median disease-free survival: 12 months vs. 18 months, *p* = 0.0012). Stage III PDAC is defined by involvement of the celiac axis, superior mesenteric artery, and/or common hepatic artery, regardless of size, or as having four or more lymph node metastases. Therefore, it is a relatively advanced cancer compared to stage I/II PDAC. Patients were categorized accordingly, and their survival rates were compared (Figure 2). Both OS and DFS rates were lower among PHC patients compared with PBTC patients (overall survival: 24 months vs. 30 months, *p* = 0.002; disease-free survival: 10 months vs. 13 months, *p* = 0.0019) with stage I/II PDAC. However, no significant difference was found between PHC patients and those with stage III PDAC (overall survival: *p* = 0.1; disease-free survival: *p* = 0.099). Figure 3 shows the comparison of survival rates of PHC and PBTC patients in the stage I/II group, without any significant difference. The survival rates were lower in PHC patients compared with PBTC patients in stage IB (30 months vs. 42 months, *p* = 0.011) and stage IIB (19 months vs. 23 months, *p* = 0.048). No significant difference between the two groups was observed for stages IA and IIA.

### 3.3. Potential Predictors for Overall Survival of PDAC Patients

Table 3 shows the factors associated with the overall survival of PDAC patients. In multivariable analysis, the OS was worse in patients with elevated CA 19-9 levels, mGPS score 1, a longer hospital stay (>15 days), complication Grade I-II, accompanying vein resection, larger tumor size, worse differentiation, higher TNM stage (stage IIB, III, IV), presence of LVI, and positive resection margin. The OS was better in patients who had received adjuvant therapy, and patients who had undergone laparoscopic or robotic surgery. Tumor location was not associated with OS in the multivariate analysis.

Appendix A show the results for the subgroup analysis according to tumor location. For PHC, the overall survival rate was worse in patients with higher mGPS (score 1), a longer hospital stay (>15 days), complication Grade I-II, accompanying vein resection, larger tumor size, worse differentiation, higher TNM stage (stage IIB, III, IV), and the presence of LVI. The OS was better in patients who had received adjuvant therapy (Appendix A). In PBTC patients, the overall survival rate was worse in patients with elevated CA 19-9 levels, a longer hospital stay (>15 days), larger tumor size, worse differentiation, higher TNM stage (stage IIB, III, IV), presence of LVI and PNI, and positive resection margin. The overall survival was better in patients who had received adjuvant therapy and those who had undergone laparoscopic or robotic surgery (Appendix A).

## 4. Discussion

In this study, the OS and DFS rates were worse in PHC patients compared with those in PBTC patients. Specifically, the OS and DFS rates in patients with stage I/II tumors, which are often diagnosed as relatively operable compared to stage III tumors, were worse in PHC patients compared with PBTC patients. When the stages were subdivided, the survival rate of PHC was consistently worse in stage IB and stage IIB patients. However, tumor location was not an independent prognostic factor. Elevated CA 19-9 levels, mGPS score 1, longer hospital stay (>15 days), complication Grade I-II, accompanying vein resection, larger tumor size, worse differentiation, higher TNM stage (stage IIB, III, IV), presence of LVI, and positive resection margin were associated with worse survival, and recent operation period, adjuvant therapy, and laparoscopic or robotic surgery were associated with better survival.

According to our findings, patients with PHC had worse survival compared with patients with PBTC. The worse prognosis of PHC could be attributed to the following reasons: First of all, the proportion of patients with LVI or PNI was higher among the PHC group. LVI or PNI, indicating aggressive tumor biology, often lead to poor prognosis [27]. Similar to our findings, several studies have reported higher N staging and higher prevalence of LVI and PNI among PHC patients [27,28,29,30]. This may imply different spread patterns and aggressiveness of cancer cells depending on primary tumor location. Another study reported poor differentiation of PHC even though the tumor size for PHC was smaller than that for PBTC [30]. Second, the different surgical procedures for PHC and PBTC may have contributed to the difference in overall survival rates. The radical antegrade modular pancreatosplenectomy (RAMPS) procedure, which is the standard operation method for PBTC, helps to achieve negative tangential margins for tumors [31]. Distal pancreatectomy, which does not require anastomosis after resection, is recognized as a safe alternative to open surgery for benign and low-malignancy tumors because laparoscopy was faster than pancreaticoduodenectomy [32,33]. For malignant tumors, no difference in postoperative complications or survival rates between PHC and PBTC have been reported [34]. Laparoscopy allows rapid post-operative recovery in PBTC patients and more efficient administration of adjuvant chemotherapy compared to open surgery. Therefore, the survival rate of these patients has improved significantly. Thirdly, PHC patients usually develop obstructive symptoms such as jaundice, which help in disease diagnosis. However, this may also delay surgery due to high bilirubin levels concomitant with poor general conditions [35]. Further, high bilirubin levels delay recovery after surgery, leading to a prolonged hospital stay. This may affect the prognosis of PHC patients as it can delay or prevent adjuvant chemotherapy after surgery. Furthermore, it is important to account for the role of micro-RNAs. Tumor location has been associated with miR-501-3p expression, which promotes carcinogenesis and recurrence of PDAC [36,37]. Low expression of miR-501-3p, associated with a lower risk of tumor recurrence, was more prevalent in PBTC patients compared with PHC patients [36,37].

AJCC TNM staging is the major prognostic factor used in clinical practice. In our data, the proportion of patients with high TNM staging was greater among the PHC group than the PBTC group. Therefore, we compared survival rates of PHC and PBTC for each stage based on the AJCC 8th edition. Stage III PDAC can be regarded as a relatively advanced cancer compared to stage I/II PDAC. Although there was no significant difference in the survival rate between the two groups in stage III PDAC, overall survival and DFS rates in stage I/II PDAC were lower in PHC patients compared to PBTC patients. When the patient group with stage I/II tumors was subdivided, PHC patients had a poorer survival rate than PBTC patients in stage IB and IIB. There was no significant difference in overall survival among patients with stage IA and IIA cancer. The 8th AJCC stage system defines stage IIB pancreatic cancer as a tumor with stage N1 regardless of stage T. The low survival rate of patients with PHC in stage IIB can be explained by the results of our multivariate analysis. Patients with lymph node metastasis and R1 resection had a worse survival rate compared to others, and PHC patients had more lymph node metastasis [38]. The difference in survival rate among patients in stage IIB can also be attributed to the difference in surgical resection range. Since PHC has many important vascular structures around it and extends to the retroperitoneum adjacent to it, it is relatively difficult to perform a clear lymph node resection. Therefore, it is difficult to obtain complete R0 resection. However, it is relatively easier to obtain a safe R0 resection because the PBTC is relatively less constrained by the surrounding structure than the PHC [39,40].

A worse survival rate for patients with stage IB PHC has also been observed in other studies. Ling et al. reported that PBTC had a lower expression of miR-501-3p compared to PHC. MiR-501-3p is known to cause the recurrence of tumors with aggressive cancer cell invasion. In the investigation of survival rate after pancreatic cancer resection, patients with metastatic disease are already excluded from surgical resection, so PBTC patients tend to have better survival than those with PHC. There was no significant difference in survival between the two groups in stage III cancer because this stage resembles a systemic disease, which includes tumor invasion to the celiac axis, superior mesenteric artery, and common hepatic artery. Other studies also did not report differences in the survival rates for this stage [21]. Wang et al. showed that tumor location of pancreatic cancer was not related to DFS in stage III pancreatic cancer patients [41]. The survival of PBTC patients in stage IB and IIB was better than the survival of PHC patients in the same stages. In fact, their survival was close to the survival of PHC patients in stages IA and IIA. Several studies have already validated TNM staging in the AJCC 8th edition; however, most of these were retrospective in design. Therefore, their results may need to be verified using the TNM staging system in a multicenter study.

Traditionally, it is known that PBTC has a worse prognosis than PHC. It is accepted that the primary reason for this is the delay in the onset of typical symptoms and final diagnosis. PHC can cause obstructive jaundice as the tumor progresses, whereas PBTC does not show symptoms until the tumor grows in size to an unresectable state. A large tumor size makes it less likely to be resected. It also increases the chances of systemic metastasis. This was demonstrated by studies that showed significantly lower survival rates for PBTCs in studies comparing 5-year survival rates regardless of resection potential [4,9,13,16,17,18,19]. However, based on the results of this study, when resection of PBTC is performed, the prognosis is better than that of PHC. Therefore, we suggest that more aggressive resection should be considered.

We confirmed that tumor location was not an independent prognostic factor. However, the findings for tumor location as a prognostic factor have varied in different studies. This suggests that there is a need to study pancreatic cancer at the bio-molecular, genetic level, to supplement clinical findings. Dreyer et al. showed that pancreatic cancer may have different molecular pathologies depending on tumor location and that PBTC genetic programs were associated with tumor invasion and poor antitumor immune responses [42]. Birnbaum et al. found differences in 334-gene expression signatures between PHC and PBTC [13]. Yin et al. analyzed differentially expressed genes and mutation signatures of PHC and PBTC [14]. It was argued that B cell and CD4+ T cell infiltration was higher in PHC, and the MATH score, which is a method of assessing intratumoral heterogeneity, was higher in PBTC compared to PHC. These findings indicate prognostic and genetic differences between PHC and PBTC. Therefore, these tumors should be regarded as substantially different pathologies that require appropriate treatment plans.

There are several limitations in this study. First, this is a retrospective study, which does not establish causality. Second, our study samples were confined to a single tertiary center, possibly resulting in selection bias. However, since the sample size was very large and consecutive, this limitation was overcome even in the subgroup analysis. Third, lesions that extended to both head and body/tail regions were excluded from the beginning, which may have influenced the overall outcome. However, the number of such cases was small.

## 5. Conclusions

The overall survival rate of pancreatic cancer patients differs according to tumor location, and PHC and PBTC have different clinical, pathological, and biological characteristics. PHC patients had a lower OS and DFS than PBTC patients for resectable PDAC. However, tumor location was not an independent prognostic factor for resectable PDAC after adjusting for potential confounders.

## Figures and Tables

**Figure 1 biomedicines-09-01706-f001:**
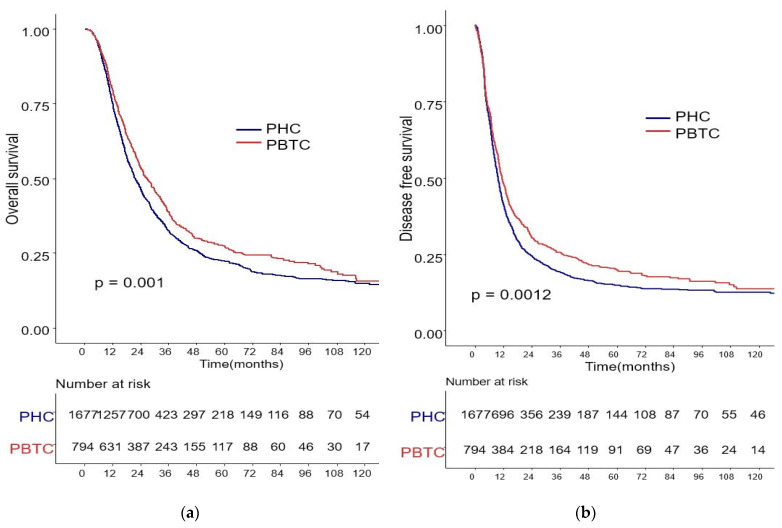
Kaplan–Meier survival curves of the cases in pancreatic head cancer (PHC) group (*n* = 1677) and pancreatic body/tail cancer (PBTC) group (*n* = 704). (**a**) The median overall survival (OS) and estimated 1-, 3-, and 5-year OS rates were 24.0 months and 75.0%, 25.2%, and 13.0% in the PHC group, and 34.0 months and 79.5%, 30.6%, and 14.7% in the PBTC group, respectively (*p* = 0.001). (**b**) The median disease-free survival (DFS) and estimated 1-, 3-, and 5-year DFS rates were 12.0 months and 41.5%, 14.3%, and 8.6% in the PHC group, and 18.0 months and 48.4%, 20.7%, and 11.5% in the PBTC group, respectively (*p* = 0.0012).

**Figure 2 biomedicines-09-01706-f002:**
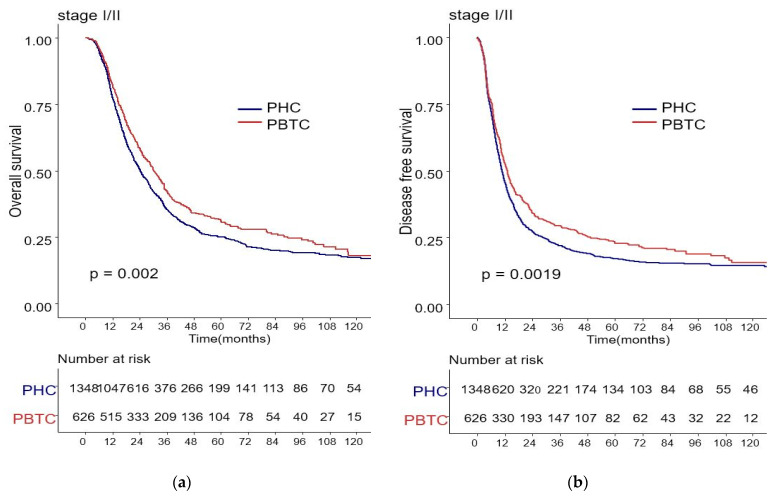
Kaplan–Meier survival curves of the cases in pancreatic head cancer (PHC) group (*n* = 1348) and pancreatic body/tail cancer (PBTC) group (*n* = 626), stratified according to cancer stage (I/II or III). (**a**) The median overall survival (OS) and estimated 1-, 3-, and 5-year OS rates for stage I/II were 24.0 months and 77.7%, 27.9%, and 14.8% in the PHC group, and 30.0 months and 82.3%, 33.4%, and 16.6% in the PBTC group, respectively (*p* = 0.002). (**b**) The median disease-free survival (DFS) and estimated 1-, 3-, and 5-year DFS rates for stage I/II were 10.0 months and 46.0%, 16.4%, and 9.9% in the PHC group, and 13.0 months and 52.7%, 23.5%, and 13.1% in the PBTC group, respectively (*p* = 0.0019). (**c**,**d**) However, no significant difference was found between PHC patients and those with stage III PDAC (overall survival: *p* = 0.1, disease-free survival: *p* = 0.099).

**Figure 3 biomedicines-09-01706-f003:**
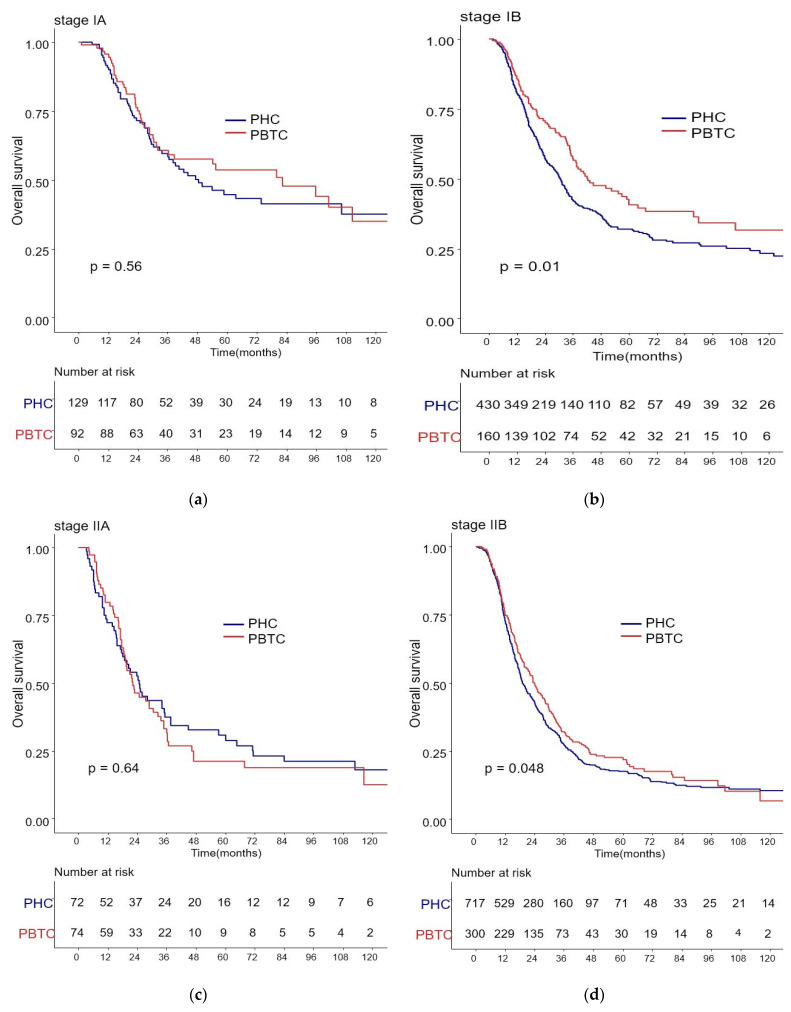
Kaplan–Meier survival curves of the cases in pancreatic head cancer (PHC) group and pancreatic body/tail cancer (PBTC) group, stratified according to the American Joint Committee on Cancer (AJCC 8th edition) Tumor-Node-Metastasis (TNM) staging system. (**b**) The survival rates were lower in PHC patients compared with PBTC patients in stage IB (30 months vs. 42 months, *p* = 0.011) and (**d**) stage IIB (19 months vs. 23 months, *p* = 0.048). (**a**,**c**) No significant difference between the two groups was observed for stages IA and IIA.

**Table 1 biomedicines-09-01706-t001:** Comparison of demographic and clinical characteristics of all pancreatic ductal adenocarcinoma patients.

		Total	PHC Patients	PBTC Patients	*p*-Value
		N (%)	N (%)	N (%)	
Age (year)	<65	1424 (57.6)	981 (58.5)	443 (55.8)	0.204
	≥65	1047 (42.4)	696 (41.5)	351 (44.2)	
Sex	Male	1465 (59.3)	675 (40.3)	331 (41.7)	0.407
	Female	1006 (40.7)	1002 (59.7)	463 (58.3)	
BMI (kg/m^2^)	<25	1912 (77.4)	1335 (79.6)	577 (72.7)	<0.001
	≥25	599 (22.6)	342 (20.4)	217 (27.3)	
ASA scores	1	54 (2.6)	33 (2.5)	21 (2.9)	0.090
	2	1881 (92.0)	1245 (93.4)	636 (89.1)	
	3	108 (5.3)	54 (4.1)	54 (7.6)	
	4	3 (0.1)	0 (0.0)	3 (0.4)	
CA 19-9	Normal	811 (32.8)	516 (31.6)	295 (38.0)	0.002
	Increased	1599 (64.7)	1118 (68.4)	481 (62.0)	
CEA	Normal	1879 (76.0)	1263 (81.6)	616 (83.0)	0.421
	Increased	410 (16.6)	284 (18.4)	126 (17.0)	
mGPS	0	2103 (85.1)	1386 (82.6)	717 (90.3)	<0.001
	1	128 (5.2)	94 (5.6)	34 (4.3)	
	2	240 (9.7)	197 (11.8)	43 (5.4)	
Period	2000–2009	722 (29.2)	514 (30.0)	208 (25.4)	0.023
	2010–2018	1749 (70.8)	1163 (69.3)	586 (73.8)	
Length of hospital stay	≤15	1287 (52.1)	767 (45.7)	520 (65.5)	<0.001
	>15	1184 (47.9)	1481 (54.3)	274 (34.5)	
POPF	No	2047 (82.8)	1484 (88.5)	563 (70.9)	<0.001
	Grade A	298 (12.1)	133 (7.9)	165 (20.8)	
	Grade B or C	126 (5.1)	60 (3.6)	66 (8.3)	
Complications	No	1595 (64.7)	1085 (64.9)	510 (64.4)	0.059
	Grade I–II	683 (27.7)	475 (28.4)	208 (26.3)	
	Grade III–IV	187 (7.6)	113 (6.8)	74 (9.3)	
Adjuvant therapy	No	707 (29.2)	492 (29.9)	215 (27.5)	0.289
	CTx	1321 (54.5)	894 (54.4)	427 (54.7)	
	CCRTx	397 (16.3)	258 (15.7)	139 (17.8)	

PHC, pancreatic head cancer; PBTC, pancreatic body/tail cancer; BMI, body mass index; ASA, American Society of Anesthesiologists; CA, carbohydrate antigen; CEA, carcinoembryonic antigen; mGPS, modified Glasgow prognostic score; POPF, postoperative pancreatic fistula; CTx, chemotherapy; CCRTx, concurrent chemoradiation therapy.

**Table 2 biomedicines-09-01706-t002:** Comparison of surgical and pathological characteristics of all pancreatic ductal adenocarcinoma patients.

		Total	PHC Patients	PBTC Patients	*p*-Value
		N (%)	N (%)	N (%)	
Operation method	Open	2065 (83.6)	1602(95.5)	463 (58.3)	<0.001
	Lap/Robot	406 (16.4)	75 (4.5)	331 (41.7)	
Vein resection	No	1828 (74.0)	1122 (66.9)	706 (88.9)	<0.001
	Yes	643 (26.0)	555 (33.1)	88 (11.1)	
Artery resection	No	2344 (94.9)	1597 (95.2)	747 (94.1)	0.227
	Yes	127 (5.1)	80 (4.8)	47 (5.9)	
Organ combined resection	No	2347 (95.0)	1643 (98.0)	704 (88.7)	<0.001
	Yes	124 (5.0)	34 (2.0)	90 (11.3)	
Tumor size (cm)	Mean (sd)	3.3 (1.7)	3.2 (1.3)	3.6 (1.9)	<0.001
Differentiation	Well	270 (10.9)	169 (10.3)	101 (13.0)	0.138
	Moderate	1832 (74.1)	1257 (76.6)	575 (74.2)	
	Poor	314 (12.7)	215 (13.1)	99 (12.8)	
T stage	T1	362 (14.7)	218 (13.0)	144 (18.1)	<0.001
	T2	1603 (64.9)	1193 (71.2)	410 (51.7)	
	T3	467 (18.9)	246 (14.7)	221 (27.9)	
	T4	37 (1.5)	19 (1.1)	18 (2.3)	
N stage	N0	996 (40.3)	642 (38.5)	354 (45.3)	0.002
	N1	1072 (43.4)	745 (44.6)	327 (41.8)	
	N2	383 (15.5)	282 (16.9)	101 (12.9)	
M stage	M0	2368 (95.8)	1626 (97.0)	742 (93.5)	<0.001
	M1	103 (4.2)	51 (3.0)	52 (6.5)	
Staging	IA	221 (9.0)	129 (7.7)	92 (11.8)	<0.001
	IB	590 (24.1)	430 (25.8)	160 (20.5)	
	IIA	146 (6.0)	72 (4.3)	74 (9.5)	
	IIB	1017 (41.5)	717 (43.0)	300 (38.4)	
	III	372 (15.2)	269 (16.1)	103 (13.2)	
	IV	103 (4.2)	51 (3.1)	52 (6.6)	
LVI	No	1163 (47.1)	718 (42.8)	445 (56.0)	<0.001
	Yes	1308 (52.9)	959 (57.2)	349 (44.0)	
PNI	No	466 (18.9)	271 (16.2)	195 (24.6)	<0.001
	Yes	2005 (81.1)	1406 (83.8)	599 (75.4)	
RM	R0	1896 (75.6)	1284 (76.6)	585 (73.7)	0.118
	R1	602 (24.4)	393 (23.4)	209 (26.3)	

Abbreviations: PHC, pancreatic head cancer; PBTC, pancreatic body/tail cancer; LVI, lymphovascular invasion; PNI, perineural invasion; RM, resection margin.

**Table 3 biomedicines-09-01706-t003:** Multivariate analysis for predictive factors of overall survival of pancreatic ductal adenocarcinoma patients.

		Univariate		Multivariate	
		HR (95% CI)		HR (95% CI)	
Location	Head	1.000		1.000	
	Body/tail	0.846 (0.766–0.935)	0.001	1.041 (0.923–1.173)	0.513
Age (yrs)	<65	1.000			
	≥65	1.086 (0.989–1.192)	0.084		
Sex	Male	1.000			
	Female	1.126 (1.025–1.237)	0.014		
BMI (kg/m^2^)	<25	1.000		1.000	
	≥25	0.799 (0.712–0.896)	0.001	0.862 (0.767–0.968)	0.012
CA 19-9	Normal	1.000		1.000	
	Increased	1.377 (1.245–1.524)	0.001	1.197 (1.080–1.325)	0.001
CEA	Normal	1.000			
	Increased	1.201 (1.061–1.359)	0.004		
mGPS	0	1.000		1.000	
	1	1.252 (1.029–1.523)	0.025	1.265 (1.038–1.541)	0.020
	2	1.290 (1.112–1.495)	0.001	1.080 (0.929–1.257)	0.315
Period	2000–2009	1.000			
	2010–2018	0.663 (0.601–0.730)	0.001		
Length of hospital stay	≤15	1.000		1.000	
	>15	1.457 (1.328–1.598)	0.001	1.266 (1.144–1.401)	0.001
POPF	No	1.000			
	Grade A	0.882 (0.766–1.017)	0.084		
	Grade B or C	0.976 (0.793–1.203)	0.823		
Complications	No	1.000		1.000	
	Grade I–II	1.179 (1.065–1.307)	0.002	1.134 (1.020–1.262)	0.021
	Grade III–IV	1.115 (0.929–1.337)	0.243	1.041 (0.863–1.256)	0.672
Adjuvant therapy	No	1.000		1.000	
	CTx	0.737 (0.664–0.819)	0.001	0.703 (0.632–0.782)	0.001
	CCRTx	0.807 (0.702–0.928)	0.003	0.633 (0.547–0.733)	0.001
Operation method	Open	1.000		1.000	
	Lap/robot	0.601 (0.522–0.692)	0.001	0.749 (0.638–0.879)	0.001
Vein resection	No	1.000		1.000	
	Yes	1.557 (1.407–1.724)	0.001	1.222 (1.096–1.363)	0.001
Artery resection	No	1.000			
	Yes	1.587 (1.315–1.916)	0.001		
Combined resection	No	1.000			
	Yes	1.138 (0.928–1.397)	0.214		
Tumor size (cm)		1.174 (1.145–1.205)	0.001	1.114 (1.078–1.151)	0.001
Differentiation	Well	1.000		1.000	
	Moderate	1.689 (1.430–1.994)	0.001	1.457 (1.236–1.718)	0.001
	Poor	2.858 (2.342–3.486)	0.001	2.393 (1.965–2.915)	0.001
T stage	T1	1.000			
	T2	1.640 (1.416–1.901)	0.001		
	T3	2.257 (1.907–2.670)	0.001		
	T4	3.052 (2.112–4.409)	0.001		
N stage	N0	1.000			
	N1	1.681 (1.514–1.866)	0.001		
	N2	2.346 (2.050–2.684)	0.001		
M stage	M0	1.000			
	M1	1.961 (1.591–2.417)	0.001		
TNM stage	IA	1.000		1.000	
	IB	1.460 (1.177–1.813)	0.001	1.104 (0.885–1.378)	0.381
	IIA	1.968 (1.508–2.568)	0.001	1.142 (0.850–1.534)	0.380
	IIB	2.424 (1.978–2.969)	0.001	1.673 (1.350–2.073)	0.001
	III	3.372 (2.705–4.204)	0.001	2.182 (1.722–2.764)	0.001
	IV	4.024 (3.043–5.321)	0.001	2.530 (1.880–3.406)	0.001
LVI	No	1.000		1.000	
	Yes	1.521 (1.386–1.670)	0.001	1.266 (1.147–1.398)	0.001
PNI	No	1.000			
	Yes	1.468 (1.295–1.664)	0.001		
RM	R0	1.00		1.000	
	R1	1.393 (1.256–1.546)	0.001	1.194 (1.066–1.337)	0.002

BMI, body mass index; CA, carbohydrate antigen; CEA, carcinoembryonic antigen; mGPS, modified Glasgow prognostic score; POPF, postoperative pancreatic fistula; CTx, chemotherapy; CCRTx, concurrent chemoradiation therapy; LVI, lymphovascular invasion; PNI, perineural invasion; RM, resection margin.

## Data Availability

The data presented in this study are available on request from the corresponding author. The data are not publicly available due to private information of patients.

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
