# Peer review of "Comparison of Characteristics and Survival Rates of Resectable Pancreatic Ductal Adenocarcinoma according to Tumor Location"

_biomedicines, 2021, doi:10.3390/biomedicines9111706_

Round 1
Reviewer 1 Report
The paper is well-written and of significant scientific interest. The study design, cohort, methods are sound, the statistical analysis methods are appropriate.
Author Response
Point 1: The paper is well-written and of significant scientific interest. The study design, cohort, methods are sound, the statistical analysis methods are appropriate. 

Response 1: Thank you for your comment. We sincerely thank you for reviewing our study and for leaving a good comment despite your busy schedule. We believe that the results of this study provide a basis for reference in future studies on pancreatic ductal adenocarcinoma according to tumor location.

Reviewer 2 Report
The authors showed PHC and PBTC have different clinical, pathological, and biological characteristics, and concluded PHC patients had a lower OS and DFS than PBTC patients. The reviewer has a few comments which the authors may address:
- Among 2,471 patients, how many cases are PHC and how many cases are PBTC? The authors need to include this information in the part of “Materials and methods”.
- According to the tables, there are 1677 cases of PHC patients and 794 cases of PBTC patients. The numbers between these two groups are too different. To compare the demographic, clinical, surgical, and pathological characteristics between these two groups, the authors are suggested to make the number of these two groups close.
- What about other characteristics of the patients? Whether these patients have other diseases such as cardiopathy etc. since these diseases can also influence the survival time. The authors need this basic information of patients in the table.
- Font size should be increased in all the figures.
Author Response
The authors showed PHC and PBTC have different clinical, pathological, and biological characteristics, and concluded PHC patients had a lower OS and DFS than PBTC patients. The reviewer has a few comments which the authors may address:
Point 1: Among 2,471 patients, how many cases are PHC and how many cases are PBTC? The authors need to include this information in the part of “Materials and methods”.

Response 1: Thank you for your insightful comment. We wrote the text “Among the 2,471 PDAC patients, 1677 (67.9%) patients had PHC, and 794 (32.1%) patients had PBTC.” In section 3.1. of results. As you pointed out, we modified that sentence in section 2.1 study participants of the materials and methods (page 2, line 30 – page 2, line 31, yellow highlight).
It is more appropriate to present the text in the materials and methods. Thank you for checking the detail.
Point 2: According to the tables, there are 1677 cases of PHC patients and 794 cases of PBTC patients. The numbers between these two groups are too different. To compare the demographic, clinical, surgical, and pathological characteristics between these two groups, the authors are suggested to make the number of these two groups close.
Response 2: Thank you for your comment. Our study was conducted as an observational study. This is because PHC and PBTC patients who underwent surgery are not randomly assigned, and the location of tumor occurrence and the stage to be confirmed after resection are determined by each patient. In addition, if it is an RCT that verifies whether a certain treatment for a specific study is effective or not, a sample size of 1:1 has high power. However, in a retrospective observational study in which all patients performed at the center were consecutively enrolled such as our study, the more patients are enrolled, the higher the power.
Before conducting the study, we considered propensity score matching as a method to match groups of patients similarly. However, although the number of both groups is similar, this method is not suitable for our study because it is a method to eliminate confounding and bias caused by patient characteristics that cause each intervention to be selected when there are two interventions. If the sample size is 1:1 by matching, we were concerned that the power of our study would be lowered in our study, which increased the power by continuously registering from 2000 to 2018.
Therefore, we conducted a retrospective observational study in which all patients were enrolled. We hope that our intentions are well understood and if the editor wishes, we will carry out the data and re-analyze it.
Point 3: What about other characteristics of the patients? Whether these patients have other diseases such as cardiopathy etc. since these diseases can also influence the survival time. The authors need this basic information of patients in the table.
Response 3: Thank you for your critical comment. Following your comment, we have conducted an additional data review of 2471 patients again. Overall survival rates, which were set as the primary outcome, were defined from the date of operation to the date of last outpatient clinic visit or death. The patient group could be divided into two groups.
The 1859 patients in the first group had an unambiguous date of death. These patients had postoperative PDAC recurrence. These were the patients who ultimately received hospice treatment or refused additional life-sustaining treatment due to the advanced stage of PDAC even though adjuvant therapy was continued. Therefore, it consisted of patients who died due to disease progression. The exact date of death was unknown for 612 patients in the second group, and survival analysis was performed based on the date of their last outpatient visit. Therefore, for patients in this group, other factors except for PDAC (such as other clinical diseases or accidents) were not included in the survival analysis. The survival analysis utilizes the partial information that “events did not occur until being censored” even in data with such censorship. In conclusion, the survival analysis shown in our study included only deaths due to the progression of PDAC. We added the following sentence to section 2.4. outcome in Materials and Methods (page 3, line 22 – page 3, line 26, yellow highlight).
We also examined the American Society of Anesthesiologists (ASA) scores to add to whether the patient had another disease. The ASA scores are a tool commonly used to classify a patient's physical fitness before surgery. The ASA scores are associated with multiple outcomes, including postoperative morbidity and mortality. Concerns about the subjectivity of ASA scores have become muted as its validity has been demonstrated in many papers. Therefore, we thought that this was a result that could be presented as basic information that shows the overall condition of the patient including the underlying disease. Therefore, after examining the ASA scores for all patients, they are presented in Table 1. Most of the patients who underwent operation corresponded to an ASA score of 2 (A patient with mild systemic disease). Patients with grade 5 (A moribund patient who is not expected to survive without the operation) who had serious factors except for PDAC that could affect survival were not included. In addition, there was no significant difference in the ASA scores between PHC group and PBTC group. We added the contents (page 2, line 48 and Table 1, yellow highlight).
Once again, we are grateful for your comment, and we hope that we have provided enough answers to your comment.
Point 4: Font size should be increased in all the figures.
Response 4: Thank you for your helpful comment. We corrected the small sizes in all the figures, and as a result, it was possible to increase the readability of the interpretation of the figure. Thank you for checking the detail.

Reviewer 3 Report
This is an interesting study on pancreatic cancer; of note, it is based on an excellent sample size, which renders very reliable the statistical analysis. Overall, this is a very good paper with a classical design. Please find here some suggestions to improve this paper:
1) please update the references and add more references, above on nodal staging (too few papers, lack of seminal papers on this topic);
2) please add histological details on the analyzed tumors: all conventional ductal adenocarcinoma? Are there any variants (e.g. adenosquamous, signet ring cell, medullary, hepatoid....and so on)
3) Please discuss the "better" prognosis of body-tail cancer than tumors in the pancreatic head. Tumors of the head usually give early symptoms if compared to body-tail cancers, thus are often diagnosed at earlier stages. The authors discussed this regarding resectable tumors, but they should discuss the whole issue, also commenting on data in the literature regarding unresectable pancreatic ductal adenocarcinoma, which represent the most important fraction of patients with this type of tumors.
Author Response
This is an interesting study on pancreatic cancer; of note, it is based on an excellent sample size, which renders very reliable the statistical analysis. Overall, this is a very good paper with a classical design. Please find here some suggestions to improve this paper:
Point 1: please update the references and add more references, above on nodal staging (too few papers, lack of seminal papers on this topic);
Response 1: Thank you for your helpful comment. Following your comment, we have written sentences “There were several studies that have suggested the anatomical location of the PDAC is a prognostic factor for survival [4-11]. However, there were recent several studies that have suggested the opposite results [12-15].” in the introduction and added references to studies on whether tumor location is a prognostic factor or not. In addition, we also found more past papers and added them to the reference (page 1, line 40 – page 1, line 43, yellow highlight). Thank you for your guidance.
Point 2: please add histological details on the analyzed tumors: all conventional ductal adenocarcinoma? Are there any variants (e.g. adenosquamous, signet ring cell, medullary, hepatoid....and so on)
Response 2: Thank you for your critical comment. Following your comment, we wrote the content in section 2.1. study participants in the Materials and Methods of the manuscript (page 2, line 17 – page 2, line 21, yellow highlight). Only patients diagnosed with PDAC were enrolled in this study. The Patients diagnosed with other histological carcinomas (such as adenosqumous carcinoma, signet ring cell carcinoma, colloid carcinoma, and so on) were very small and generally had a poorer prognosis than PDAC. Therefore, we excluded those with other histological diagnoses. By improving the points, we were able to convey the meaning more clearly to the readers. Thank you for your guidance.
Point 3: Please discuss the "better" prognosis of body-tail cancer than tumors in the pancreatic head. Tumors of the head usually give early symptoms if compared to body-tail cancers, thus are often diagnosed at earlier stages. The authors discussed this regarding resectable tumors, but they should discuss the whole issue, also commenting on data in the literature regarding unresectable pancreatic ductal adenocarcinoma, which represent the most important fraction of patients with this type of tumors.
Response 3: Thank you for your important comment. Traditionally, PBTC has a worse prognosis than PHC. PHC can cause obstructive jaundice as the tumor progresses, whereas PBTC does not show symptoms until the tumor grows in size to unresectable state. A large tumor size makes it less likely to be resected. It also increases the chances of systemic metastasis. This was shown by studies (reference 4,9,13,16-19) that showed significantly lower survival rates for PBTCs in studies comparing 5-year survival rates regardless of resection potential. However, based on the results of this study, when resection of PBTC is performed, the prognosis is better than that of PHC. Therefore, we suggest that more aggressive resection should be considered.
We added the content in the discussion so that readers do not misunderstand (page 11, line 34 – page 11, line 41, yellow highlight). By improving the points, we were able to convey the meaning more clearly to the readers. Thank you for your guidance.

Reviewer 4 Report
The paper is well written with evaluation of sufficient number of patients who underwent pancreatectomy for PDAC. However, this is fully a clinical study and I do not think the paper matches the scope and aims of the Journal. The paper should be submitted to a journal dealing with clinical medicine.
Author Response
Point 1: The paper is well written with evaluation of sufficient number of patients who underwent pancreatectomy for PDAC. However, this is fully a clinical study and I do not think the paper matches the scope and aims of the Journal. The paper should be submitted to a journal dealing with clinical medicine.

Response 1: Thank you for your comment and for taking the time to review our paper. Thank you for complementing that our paper is well written with an evaluation of the sufficient number of patients who underwent pancreatectomy for PDAC. We believe that our paper research is of interest to your journal and to international readers (Biomedicines, Special Issue “Pancreatic Cancer: From Mechanisms to Therapeutic Approaches) because our study showed the prognostic factor of the pancreatic tumor location and survival outcomes between the location of PDAC using real consecutive data.
Although several past studies have suggested that the anatomical location of pancreatic tumors is a potential prognostic factor for survival, the prognostic significance of tumor location is still controversial. We found that this study showed that the location of the tumor itself is not a potential prognostic factor through multivariable analysis. Therefore, we show the clinicopathological characteristics of disparity between PHC and PBTC. We think that based on these characteristics, it can serve as a basis for progressing in the direction of the genetic and molecular characteristics of which may be different.
Some studies have reported that PBTC has a worse prognosis than PHC. This is considered to be due to the fact that, due to the late clinical onset, the stage is often advanced at the time of diagnosis and the actual resection rate is low. Therefore, we wanted to investigate whether there is a difference in the prognosis of PBTC and PHC in the case of being resected. Although it is the result of a single-center, we provided many patient data from 2000 to increase the power of the test. In addition, as a study in which a sufficient number of surgeons with a lot of clinical experience participated, we think that the study related to this subject presented a meaningful result.
Therefore, we believe that the results of this study provide a basis for reference in future studies on pancreatic cancer according to tumor location. Although there is no future specific treatment, we believe that it has implications for the topic of the pancreas cancer special issue. Thanks again for your comments, and we hope your comments have been adequately answered.

Round 2
Reviewer 2 Report
In the revised article, the authors modified the manuscript and figures referred to the comments, answered the questions comprehensively.
Reviewer 4 Report
The authors adequately responded to reviewer's comment. I do not have any additional comments for further revision.